# Social Determinants of Health Influencing the New Zealand COVID-19 Response and Recovery: A Scoping Review and Causal Loop Diagram

**Sudesh Sharma** *, **Mat Walton** and **Suzanne Manning**

Institute of Environmental Science and Research, Porirua 5022, New Zealand; mathew.walton@esr.cri.nz (M.W.); Suzanne.manning@esr.cri.nz (S.M.)
* Correspondence: sudesh.sharma@esr.cri.nz

**Abstract:** The Coronavirus pandemic of 2019–20 (COVID-19) affected multiple social determinants of health (SDH) across the globe, including in New Zealand, exacerbating health inequities. Understanding these system dynamics can support decision making for the pandemic response and recovery measures. This study combined a scoping review with a causal loop diagram to further understanding of the connections between SDH, pandemic measures, and both short- and long-term outcomes in New Zealand. The causal loop diagram showed the reinforcing nature of structural SDH, such as colonization and socio-economic influences, on health inequities. While balancing actions taken by government eliminated COVID-19, the diagram showed that existing structural SDH inequities could increase health inequities in the longer term, unless the opportunity is taken for socio-economic policies to be reset. Such policy resets would be difficult to implement, as they are at odds with the current socio-economic system. The causal loop diagram highlighted that SDH significantly influenced the dynamics of the COVID-19 impact and response, pointing to a need for purposeful systemic action to disrupt the reinforcing loops which increase health inequities over time. This will require strong systems leadership, and coordination between policy makers and implementation at local level.

**Keywords:** COVID-19; social determinants; health system; New Zealand

## 1. Introduction

The infectious disease caused by the coronavirus SARS-CoV-2 triggered a global pandemic in 2020 (henceforth 'COVID-19') with profound consequences for millions of people. Countries responded to the COVID-19 pandemic in a variety of ways, with equally variable effectiveness. New Zealand, also known as Aotearoa (indigenous Maori name), was relatively successful in controlling the disease, aided by its comparative isolation from the rest of the world. When community transmission was first detected in March 2020, the government implemented strict measures designed to eliminate the virus, including a stringent lockdown with most of the population staying at home for six weeks. During this period there were 1503 cases of COVID-19, 95 hospital admissions and 22 deaths [1]. Measures were relaxed after this first wave, and then less stringent measures were temporarily applied on a regional basis when an outbreak occurred in August. By 16 December 2020 there had been 2100 cases and 25 deaths, and most restrictions on movement, business operations and social gatherings had been lifted, while borders remained tightly controlled (https://nzCOVIDdashboard.esr.cri.nz, accessed on 16 December 2020). Since that time there have been several other, smaller outbreaks, with more regionally focused response measures applied.

While the direct impact of COVID-19 in terms of cases and deaths has been huge globally, the indirect impacts from the responses to the pandemic were also significant. The social determinants of health (SDH) conceptual framework illustrates how determinants

such as income, gender, ethnicity, education and occupation influence an individual's socio-economic position, and subsequently health outcomes [2–4]. The critical link between socio-economic position with health was shown after the financial crisis of 2008 [5], and the COVID-19 pandemic may have similar impacts. Bambra and colleagues have critically discussed the potential impact of COVID-19 on existing socio-economic and environmental determinants of health, and ultimately on health equity [6]. Systems science can help understanding of relationships between micro and macro determinants of health from which population health and health inequalities are produced [7]

Within New Zealand, health inequities have been regularly shown across communicable and non-communicable disease, access to health care, and SDH [8–11]. Health inequities are recognized by government and are an active area of policy and health services e.g., [12–14]. At early stages of COVID-19, predictions in New Zealand were that, after the elderly, Māori and Pacific communities were most at risk of high morbidity because of the compounded effects of underlying health conditions, socio-economic disadvantage and structural racism [15–17]. International literature shared concerns that populations with lower socio-economic status would be disproportionately impacted by COVID-19 response measures in the longer term, based on historical precedents and the link between SDH and intergenerational disadvantage [18–23].

The direct impact of COVID-19 was relatively small in New Zealand because of the successful elimination strategy [24]. There were no significant differences between ethnic groups in severity of cases resulting from the first, and largest, community outbreak [1,17]. The subsequent community outbreaks, with short, regional restrictive lockdowns, were concentrated in economically disadvantaged geographic areas.

Evidence to date about indirect impacts of COVID-19 on SDH appears more ambiguous. Rates of gambling have reportedly dropped, while drinking alcohol and smoking increased during lockdown, with smoking rates remaining high particularly amongst Māori [25]. The rise in unemployment rates across 2020 were modest yet unequal, with higher rates experienced by women, younger people, Māori and within some regions [26]. Unemployment and rising housing costs were two major causes suggested for the increased number of food parcels delivered by social support agencies over 2020 [26].

When the restrictive lock-down started in March 2020, the government provided support payments: a wage subsidy to employers, a permanent increase in all income support and a temporary increase in winter energy support. However, a June 2020 survey reported that a third of respondents had seen a reduction in income as a result of COVID-19, with almost a fifth reporting they did not have enough money to meet usual daily needs [25]. Pacific community respondents were most likely to report not having enough money for every day needs, and symptoms of depression or anxiety were highest amongst the group that had seen a loss of income [25]. A second survey conducted in August/September 2020, with people who had received some form of income support payment, reported 26% of respondents experienced a significant drop in personal income since March 2020, and many reported difficulty meeting household costs [27].

Positive impacts on health were also noted from lockdown periods. Huang and colleagues identified a large decline in influenza and other viral respiratory infections across multiple surveillance types [28]. Māori and community led responses to supporting community members during lockdown demonstrated new ways of collaborating within communities and between central government and communities [17,29]. Long term impacts of past financial crises have been shown on SDH and health outcomes [5]. Recovery from COVID-19 pandemic situation may seek to utilize lessons from positive impacts identified to date, whilst addressing drivers of health related to SDH.

The COVID-19 experience to date in New Zealand of direct and indirect impacts suggest complex emergent health and equity effects generated through interaction of many influences on health and health equity. To support government action in COVID-19 response and recovery, understanding how the structure of the system contributes to health and equity outcomes is important. The aim of this study is to support analysis of

government policy and programme decisions through insights generated from a qualitative causal loop diagram. This study sits alongside other epidemiology and qualitative studies within a larger Co-Search project (https://www.otago.ac.nz/wellington/departments/publichealth/research/heiru/co-search/index.html accessed on 5 June 2021), focused on supporting equity in the pandemic response and recovery. The causal loop diagram developed here has two direct uses. The first was to provide a systems and health equity lens through which to view COVID-19 response and recovery actions, and help consider upcoming priorities for action. The second purpose was as an input into group model building workshops to further explore dynamics between SDH and COVID-19 at a local community level.

The method section presents the process followed in undertaking a scoping literature review from which a causal loop diagram was developed. The final section uses the causal loop diagram as a lens to consider a selection of COVID-19 response and recovery actions within New Zealand over 2020 as an example of applying a systems and health equity lens. Finally, there is a discussion of the implications of system insights for the ongoing pandemic recovery, including reflection on causal loop diagramming as a tool.

## 2. Materials and Methods

The study framework was adapted from Arksey and O'Malley [30], who describe the purpose of a scoping review as mapping the breadth of relevant literature in a particular field. The value of a scoping study is that it can quickly provide a description of available research which may be useful for policy makers or provide the basis for ongoing work. The study followed the process of defining research questions, identifying and selecting relevant studies, analysis and visualization using a causal loop diagram.

### 2.1. Defining the Research Questions

The Co-search project focuses on equity in the COVID-19 response and recovery in New Zealand, especially for Māori and Pacific communities who have historically experienced discrimination and disadvantage. The COVID-19 pandemic has presented both a challenge and an opportunity: on the one hand the pandemic was predicted to exacerbate inequities and disproportionately affect disadvantaged communities [15], yet on the other hand there is an ongoing opportunity to explicitly and proactively design and continually adapt responses to improve equity in health outcomes. This scoping study is a component of the overall project that frames the complex problem as a system to be analyzed using systems approaches from a SDH perspective.

After discussion amongst the research team, two questions were established that guided the review:

(i) what are the social determinants of health and equity in the context of COVID-19 in New Zealand?

(ii) how are these social determinants of health and equity interacting and affecting the COVID-19 response and recovery decisions and actions?

### 2.2. Identifying and Selecting Relevant Literature

A preliminary screening list of relevant published literature was produced by searching the PubMed database using key terms: [COVID-19 OR corona virus OR pandemic] AND [determinants OR social determinants OR inequity OR Māori OR inequality OR equity OR equality OR public health OR health promotion OR health policy OR vulnerable] AND [New Zealand OR Australia OR United States OR Canada], between January and July 2020. In addition, grey literature such as government reports, policy documents and expert commentaries, were found through searches of Google and government ministry websites. Search terms were adjusted [30], to ensure that a wide range of social determinants were covered and that Māori and Pacific peoples were well represented. The list included about 161 pieces of international literature, and 86 pieces of literature from New Zealand.

Two researchers jointly made decisions on the final list based on the abstracts, where the main criterion was the potential of the literature to address the research questions while considering the short time frame. The final selection included 30 pieces of literature, more than half from New Zealand (see supplementary Figure S1 for study selection flowchart).

### 2.3. Analyzing the Data

Literature was coded by the researchers in Dedoose (qualitative data analysis software), using both a high-level coding tree based on the WHO SDH framework and deductive codes to capture emerging themes. The researchers held regular meetings to refine the codes and discuss emerging themes.

In parallel, a spreadsheet was developed to capture variables and causal linkages for developing a causal loop diagram. For each piece of literature, one or more causal variables were identified along with the accompanying impacts, responses, suggested linkage chains between variables and the direction of each linkage. Multiple causal chain linkages were sometimes suggested, to show alternative interactions between the variables. The causal loop diagram that was developed from this spreadsheet is therefore only one potential way to show the system.

### 2.4. Summarizing through Causal Loop Diagram

The causal loop diagram was developed using the conventions of system dynamics [31]. A causal loop diagram is a system dynamics tool that maps the relationship between different elements or variables within a system and visualises the feedback structures. It consists of variables connected with arrows and polarities (+ or −) which denote causal influence (mechanism) and direction of influence (increasing or decreasing) [31]. A + sign by the arrowhead between two elements (for example, A $\rightarrow^+$ B) indicates that a change in A causes a change in B in the same direction. A – sign between two elements indicate that an increase in one variable causes a decrease in the second, or vice versa, i.e., changes in the in the opposite direction. However, care should be taken in interpreting these polarities as sometimes the link polarity may not be straightforward to interpret. For example, a positive link between infection rate and sick population does not mean decrease in infection rate leads to decrease in sick population, but sick population would be less compared to what would otherwise be, i.e., sick population will continue to increase but at a slower rate. When there is a delay between the cause and the subsequent effect, this is indicated by // signs in the middle of the arrow. Further, when the link is based on explicit mechanisms discussed in the literature, the arrow is in black and includes literature reference numbers (see Table 1 in results). When the link is inferred, based on overall logic and known contexts, it is shown in red. These links should be interpreted cautiously but are essential to illustrate how feedback loops could be operating.

A feedback loop is created when a causal chain, from variable to variable, can be traced back to its originating variable. These loops are either balancing (indicated by a 'B' in the middle of the loop in the diagram) or reinforcing (indicated by a 'R'). A balancing loop acts to stabilise a variable, so that when the variable changes, the feedback loop acts to reverse the change. In contrast, reinforcing loops involve a chain of actions that amplify the original change in the variable, producing a cycle of continuous growth or decline. A causal loop diagram has multiple interacting feedback loops and making sense of these interactions helps to interpret the potential dynamics and system behaviour around problem of interest [32–34]. The analysis and sense making of the diagram was guided by the WHO SDH framework.

The causal loop diagram is a model of a system, and like all models, is a partial and incomplete reflection of reality; yet visualising a system using this tool can provide useful insights about how different parts of a system may be interacting to cause patterns over time. Further, the diagram may have potential biases due to addition of some inferred linkages and interactions not directly identified in the literature.

**Table 1.** Description of the papers and causal mechanism discussed within those papers.

| ID | Paper | Focus Country | Social Determinants of Health | Model Variables and Connections |
|---|---|---|---|---|
| 1 | Abrams et al. | US | Poverty, homelessness, housing, ethnicity, smoking, pre-existing health conditions, physical distancing, COVID-19 morbidity, socio-economic impact | Colonization—SE inequities; SE inequities—Infection risk—Cases—SE impact—SE inequities; Consequences—Health access |
| 2 | Al-Bausaidi et al. | New Zealand | NZ lockdown, quarantine requirements, pre-existing health conditions, digital divide, primary care services, access to medicines | Health response—Cases; Health response—Infection risk |
| 3 | Arnold et al. | Australia and New Zealand | Age, indigenous people, rural, pre-existing conditions, COVID-19 morbidity, primary care services, resource management | Health access—Infection risk—Cases; Health access—SE inequities; Service design—Health access |
| 4 | Azar et al. | US | Racism, health service user experience, health seeking behavior, COVID-19 infection, COVID-19 hospitalization rate | Colonization—Services design—Health response/Indigeneity/Health access—Infection risk—Cases |
| 5 | Bandyopadhyay et al. | New Zealand | NZ pandemic response; mental stress; community cohesion; | Health responses—Benefits/Consequences/Cases |
| 6 | Boston | New Zealand | Linear economy; ecological crisis; policy reset opportunity; circular economy; NZ/global pandemic responses; fiscal recovery packages | Infection risk—SE impact; Govt support—SE impact/Capitalism/Benefits—Capitalism |
| 7 | Beland et al. | Canada | Canada federal & provincial health system; underfunding/low priority; aged care facilities; staffing, processes; age; COVID-19 morbidity | Service design—Health access—Infection risk |
| 8 | Carr | New Zealand | Tourism; indigenous businesses, racism, colonization, socio-economic status/impact, travel restrictions | Colonization—SE inequities—Infection risk—SE impact—SE inequities; Infection risk—Cases |
| 9 | Crotty et al. | Australia | Australia health care system; policy and funding settings; aged care facilities; staffing, processes; older age. vulnerable people; high morbidity | Service design—Health access—Infection risk |
| 10 | Doogan et al. | Global | Political leadership; information & communication; country pandemic response measures; compliance rate; emotional appeal | Leadership—Health response—Cases/Infection risk |
| 11 | Fitzgerald et al. | Global | Young age; morbidity; preparedness; centralized response; compliance rate; virus containment; prevention measures | Leadership—Health response—Cases/Infection risk |
| 12 | Fletcher | New Zealand | Poverty; welfare policies; inequalities; access; COVID-19 risk | SE inequities—Family support—SE inequities; Service design—SE inequities/Health access—Infection risk |

**Table 1.** *Cont.*

| ID | Paper | Focus Country | Social Determinants of Health | Model Variables and Connections |
|---|---|---|---|---|
| 13 | Foley et al. | Australia and New Zealand | Young age; health needs; pediatric physicians; information; leadership; health system capacity/resilience; preparedness for pandemics | Service design—Health response—Infection risk/Cases |
| 14 | Furlong et al. | Australia | Structural, historical racism and colonization; Asian population discrimination; social capital and harmony; indigenous people; culture; economic and social disadvantage; co-existing health conditions; rural location; tobacco consumption; mental health resilience | Colonization—SE inequities—Infection risk; Leadership—Health response; Consequences—Health access |
| 15 | Galea-Singer et al. | New Zealand | Physical distancing; substance misuse therapy; virtual therapy clinics; research gaps | Service design—Health access—SE inequities |
| 16 | Hamill et al. | New Zealand | WHO declaration; global response; NZ response; lockdown and travel restriction; accidents and trauma rate among children | Health response—Benefits |
| 17 | Hawkins | US | Poverty; racism; vulnerability; occupational status; essential worker; job security and entitlements, at risk groups; poverty cycle | SE inequities—Family support—SE inequities; SE inequities—Infection risk—SE impact—SE inequities; Infection risk—Cases |
| 18 | Junior et al. | Global | Rural location; socio-economic condition, colonization and historical trauma; Western intervention; access to mental health services; information; health workers limited availability; indigenous mental health status and access to services; reinforcing vulnerability | Colonization—Service design—indigeneity—Health access—Infection risk—SE Impact/Cases |
| 19 | Kokaua | New Zealand | Race/ethnicity; vulnerability; COVID-19 advocacy; society development; systemic bias; cultural measures; inequity in health sector | Colonization—Service design—indigeneity—Health access—Infection risk—Cases |
| 20 | Laster Pirtle | US | Structural racism; historical trauma; socio-economic disadvantage; racial capitalism; health inequity; COVID-19 risk and vulnerability | Colonization—Service design—Health access/SE inequities—Infection risk—Cases |
| 21 | Laurencin et al. | US | Historical racism, poverty, crowded housing, limited data, misinformation; pre-existing social & health inequity; limited access; design of health system; disproportionate impact on disadvantaged groups | Colonization—Service design—Health access—SE inequities |

**Table 1.** *Cont.*

| ID | Paper | Focus Country | Social Determinants of Health | Model Variables and Connections |
|---|---|---|---|---|
| 22 | Levin | US | Faith and religion context; scientific divide; trust; misinformation; COVID cases; faith based medical centers; collaboration and coordination with religious agencies | Indigeneity—Health access—SE inequities—SE impact |
| 23 | McMeeking et al. | New Zealand | Structural racism; historical inequity; underlying health conditions; Māori collective and cultural capital; government response; Māori empowerment and ownership; trust; access to services and information; at risk population; socio-economic status | Colonization—Service design—Indigeneity/Health access/Health response—Indigeneity—Community response/Health access—Infection risk; SE impact—SE inequities; SE impact—Innovation—SE impact; SE impact—Community response—Health access |
| 24 | Ministry of Education | New Zealand | Essential workers, low-income bracket; sick leave and flexibility at work; vulnerable age; ability to work from home; gender and essential work; socio-economic impact | SE inequities—Infection risk |
| 25 | SocialLink | New Zealand | Lockdown and travel restrictions; social services disruption; extra workload; increased travel expenses; effect on fundraising; Impact on livelihood and mental health; violence against women and children; | Health response—Resources/Infection risk/SE impact—Community response |
| 26 | St-Denis | Canada | Age; gender; essential work status; education status; poverty status; risk of COVID, reinforcing socio-economic condition | SE inequities—Family support—SE inequities; SE inequities—Infection risk—SE impact/Cases |
| 27 | Steyn et al. | New Zealand | Age; race/ethnicity; socio-economic status; structural racism; crowded living spaces; access to health services; rural location; at risk group; COVID cases | SE inequities—Family support—SE inequities; SE inequities—Infection risk—Cases; Colonization—Service design—Health access—SE inequities |
| 28 | Wilson et al. | New Zealand | Design of health system and infrastructure; health system gaps, preparedness; pre-existing inequities; health protection workforce; pandemic response strategies; precautionary principles; unintended benefits; socio-economic impact; infection rate; green reset opportunities; reduction in related harms | Service design—Indigeneity/Health response—Indigeneity; Health response—SE impact; Health response—Benefits; Leadership—Benefits/Resources; Cases—Leadership/Consequences; Govt support—SE impact/Benefits |
| 29 | Yashadhana et al. | Australia | Structural racism; health status, inequities; access to and design of health services; socio-economic status; trust; utilisation; funding of indigenous services; at risk group; COVID-19 cases | Colonization—Service design—Health access/Indigeneity/Health response—Indigeneity—Health access—SE impact/Infection risk; Consequences—Health access |
| 30 | Anderson et al. | New Zealand | Pre-existing social conditions; violence; poverty; young age; pandemic responses; child development; employment status; pre-existing social conditions | SE inequities—Family support—SE inequities; |

## 3. Results

### 3.1. Characteristics of the Literature

Thirty pieces of literature were selected, 16 from New Zealand alone or New Zealand and Australia combined [15–17,35–47], 3 from Australia alone [19,21,48], 2 from Canada [20,49], 6 from the United States [18,22,23,50–52], and 3 with a more global perspective [53–55].

The type of literature included peer reviewed original research [17–21,23,38,41,42,47, 49,51–54], commentary [22,37,39,43,44,48,50,55] and non-peer reviewed reports [15,16,35, 36,40,45,46]. The literature was taken from a wide range of journals and websites, mostly only one article or report from each source except for 4 articles from a special COVID-19 issue of *The Policy Quarterly* [17,38,41,47], a New Zealand journal aimed at government policy makers.

The range of social determinants included an emphasis on the health system [16, 20,23,35,39,40,45,47,49,53], society and culture [17,19,38,43,52,54,55], economy [37,46,48], environment [18] and education [50]. Some articles emphasized multiple social determinants [15,21,22,36,41,42,44,51].

Table 1 lists the papers reviewed and includes the initial notes on SDH identified by authors from within these papers (column 4), and causal mechanisms (not shown) which were further refined and used to develop the causal loop diagram. The last column lists the final variables and connections contained in the model.

### 3.2. Causal Loop Diagram

The causal loop diagram (Figure 1) presents a way of visualizing the interactions and interconnections of various SDH influencing the COVID-19 response and recovery, as discussed in the selected literature. The causal loop diagram shows four broad categories: structural determinants (colored orange), health system determinants (colored green), health and socio-economic impact (colored yellow); socio-economic and health response (colored blue). In the causal loop diagram, causal influences are indicated by the arrows between variables, with the ID of the literature that shows this link.

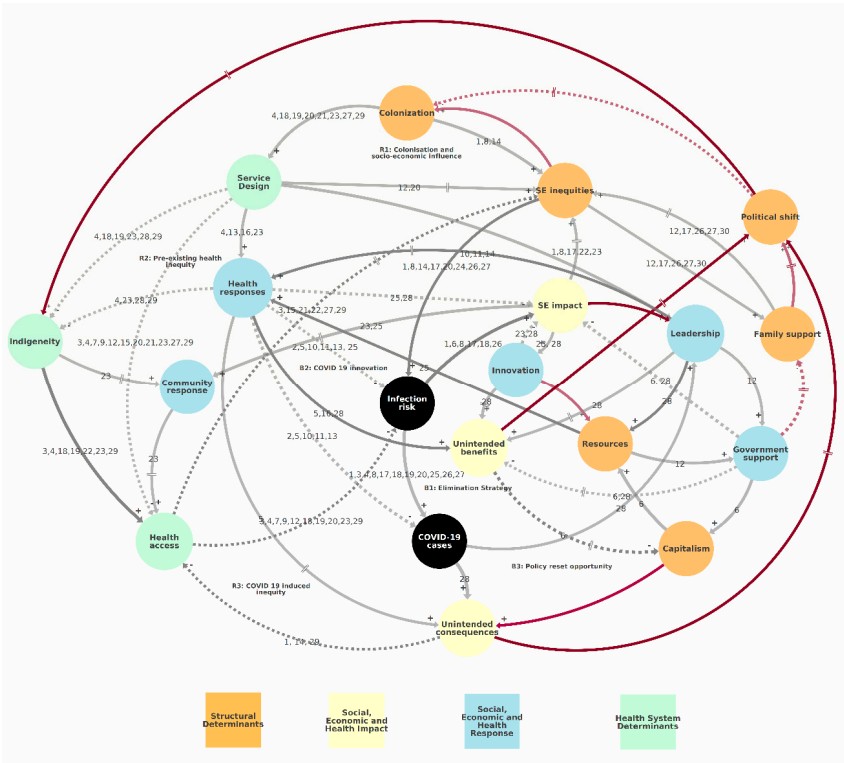

**Figure 1.** Causal loop diagram of the social determinants of health and their interaction with COVID-19 response and recovery.

Three reinforcing loops (R) can be identified in this causal loop diagram, which act to increase or decrease inequities. These loops have been labelled as *R1: Colonization and socio-economic influence*, *R2: Pre-existing health inequities* and *R3: COVID-19 induced inequities*. Three balancing loops (B) are also shown, which act to balance or improve current health outcomes: *B1: Elimination strategy*, *B2: COVID-19 innovation*, and *B3: Policy reset opportunity* (Table 2; see Supplementary Figure S2 for individual figures of loops).

**Table 2.** Description of the papers and causal mechanism discussed within those paper.

| ID | Name | Type | Causal Loop/s in the Model |
|---|---|---|---|
| R1 | Colonization and socio-economic influence | Reinforcing loops | Colonization → Service design → SE inequities (→ Family support) → Colonization |
| R2 | Pre-existing health inequity | Reinforcing loops | Colonization → Service design (→− Indigeneity →) →− Health access →− SE inequities → Colonization |
| R3 | COVID-19 induced inequity | Reinforcing loops | Health access (→− SE inequities →) →− Infection risk (→ SE impact → SE inequities) → COVID-19 cases→ Unintended consequences →− Health access |
| B1 | Elimination Strategy | Balancing loops | COVID-19 cases → Leadership → Health response (→− Infection risk →) →− COVID-19 cases<br><br>SE Impact → Leadership → Resources (→ Health responses →− Infection risk → SE impact) → Government support →− SE impact |
| B2 | COVID-19 innovation | Balancing loops | Health access (→− SE inequities →) →−Infection risk → SE impact (→ Innovation →−) → Community response → Health access |
| B3 | Policy reset opportunity | Balancing loops | Infection risk → SE impact → Leadership (→ Resources) → Health response → Unintended benefits → Political shift → Indigeneity → Health access (→− SE inequities →) →− Infection risk<br><br>Infection risk → SE impact → Leadership (→ Resources) → Health response → Unintended benefits →− Capitalism→ Unintended consequences→− Health access (→− SE inequities →) →− Infection risk |

Note: Elements within brackets along the causal loops show alternative linkage with immediately preceding and/or following elements in Figure 1.

Structural determinants of health (orange) include contextual factors such as *colonization*, *service design*, *socio-economic inequities*, *capitalism*, *political shift* (a shift in political mindset and subsequent policies towards equitable society) and *resources*. *Socio-economic inequities* are shaped by long term historical forces such as *colonization* and the resulting structural racism, and short-term fluctuations such as unemployment due to COVID-19. *Colonization* has negatively impacted the socio-economic status of the indigenous Māori population, although there is variation within that group (R1). The diagram suggests that *socio-economic inequities* influences the morbidity and mortality rates for COVID-19 (*COVID-19 cases*) as the reinforcing loops R2 and R3 suggest that those on lower incomes, with insecure jobs, poor quality housing and difficulty in affording essentials such as food and electricity, are more at risk from COVID-19 (*infection risk*). In the causal loop diagram, Western science principles and values are seen as dominating, marginalizing the other perspectives on maintaining wellbeing, including indigenous Māori approaches, contributing to underutilization of health services and inequitable *health access* (R2). This in turn would reinforce health and socio-economic inequities and further embed structural discrimination (R1, R2, R3).

The first balancing loop is B1: Elimination strategy. The literature identified science-based strategies with politicians, bureaucrats and scientists working together to eliminate

the COVID-19 virus from the country. Part of B1 is the level of public compliance with public health measures (e.g., staying at home when asked), which could be impacted by misinformation, or the 'infodemic' as described in some literature.

Balancing loop two (B2: COVID-19 innovation) suggests that actions to "do things differently" in face of COVID-19 could limit the dynamics resulting from reinforcing loops that tend to maintain or increase health inequities over time. Examples include community led responses to supporting vulnerable people in their homes with food and social support. These different approaches could mitigate some short term adverse socio-economic impacts of the COVID-19 measures such as the lockdown.

The literature described several consequences—both negative (*unintended consequences*) and beneficial (*unintended benefits*)—to COVID-19 responses. The systems diagram shows some *unintended benefits* such an increase in cycling rates, reduction in air and water pollution, and job innovations. Some literature describes COVID-19 as providing an opportunity for a 'policy reset' (B3: Policy reset opportunity), with commentators putting forward ideas of how policy could be changed to improve the environment along with health and social inequities [17,38,39,45,47,49]. However, the diagram also identifies resistance leading to delays in the policy reset due to current socio-economic system (R1).

## 4. Discussion

This study sought to identify social determinants of health and equity (SDH) in the context of COVID-19 in New Zealand, and how these SDH may interact and affect the COVID-19 response and recovery. A scoping literature review was undertaken from which a causal loop diagram was developed. This approach of systems mapping enabled consideration of interactions and possible causal mechanisms of the key SDH impacting COVID-19 response and recovery.

Using systems perspectives to help describe and understand complex public health issues has received some attention in recent years, including for COVID-19 [56–59]. Causal loop diagramming is a tool that can help structure and visualize the SDH variables and their causal relationships as an interconnected system, for the purpose of generating insights. Such insights can be used to help consider ways that the structure of the system might act to help or hinder interventions, and indeed how interventions may act to help or hinder each other.

Key insights from this causal loop diagram suggest continued health inequities experienced by those with lower level of resources across SDH. Such resources are structured through numerous systems (i.e., employment, education, justice) that privilege certain worldviews and concentrate political power [60,61]. The analysis suggested that inequities are likely to get worse without radical change. The causal loop diagram also suggests that inequities in health service provision are likely to continue in design of COVID-19 response and recovery, unless purposefully avoided.

Another insight identified from the causal loop diagram relates to the long timeframe of causal mechanisms (e.g., *colonization*) and the delays within reinforcing loops, compared to relatively shorter time periods within balancing loops. Here, actions that positively support health in the shorter term may also support longer term negative trends within reinforcing loops.

As an example, let us consider government income support payments in New Zealand. It has often been acknowledged that income support payment levels are too low for supporting wellbeing, with a focus on incentivizing work over wellbeing and meaningful engagement in society whilst receiving income support. The government-appointed Welfare Expert Advisory Group recommended an increase in all categories of income support, including over $100 per week in several categories [62]. As part of the COVID-19 response, the government permanently increased all income support payments by $25 a week; doubled the winter energy payment for 2020; and introduced a COVID-19 income relief payment for a period of 12 weeks for those who lost their job due to COVID-19, which was set at a much higher payment than the usual unemployment payment [27]. Using the

causal loop diagram as an analytical lens, we could view the government income support changes as part of the COVID-19 innovation balancing loop (B2), mitigating some expected negative impacts on SDH. We could also consider the changes as continuing support for low levels of income support payments that prioritizes work incentives and supports distinction between deserving and underserving poor, because those who lost jobs due to COVID-19 were being treated differently to those who lost jobs prior to COVID-19 (R1). On this analysis, we could expect pre-COVID-19 patterns of inequity between those in-work and those on income support payments to continue longer term, even though the government invested significantly on supporting incomes during 2020.

Another example could be the COVID-19 vaccination plan. Early criticism of Government response from Māori clinicians and academics resulted in on-going changes to Government response. More recently in 2021, plans for vaccine roll-outs have specifically supported access for Māori and Pacific Island groups through resourcing of Māori and Pacific-led health providers (B3). At the same time, vaccine plans have been criticized as likely to reinforce existing inequities (R2), for example through population wide age cut-offs for those considered more vulnerable to COVID-19, that do not take into account different experiences of morbidity within sub-population groups [63].

While the causal loop diagram suggests continuing health inequities, it shows that balancing loops can over time influence reinforcing loops, and suggests opportunities to reduce health inequities through a policy 'reset'. Government responses have shown the ability to do things differently, including additional income support payments discussed above and direct resourcing to support Māori-led community responses (within Figure 1 *indigeneity → community response → health access*). Feedback loops that currently create increasing inequity also provide mechanisms for reducing inequity. For example, marginalization of Māori perspectives in health service design has been identified as influencing lower rates of health service utilization and poorer health outcomes for Māori. However, the same causal linkages would suggest that an approach that gave equal weighting, or privileged, Māori perspectives would increase utilization and improve health outcomes in a reinforcing spiral. Likewise, the ability for recovery to support biodiversity and climate goals has been noted, but would require a shift in economic paradigm [38]. The challenge, well discussed within systems literature, is that changing worldviews and paradigms that underpin how a system is structured are likely to have biggest impact on outcomes generated by that system, yet are the hardest change to achieve [64–66].

How can underlying worldview and paradigms be transformed to orient systems to reduce health inequities in COVID-19 response and recovery? The concept of systemic leadership likely provides a partial answer. Systemic leaders act with knowledge of how the system is interacting with SDH, understand how current health outcomes are shaped by history, and create space for including diverse paradigms and perspectives. System leadership fosters collective leadership, shared vision, collective action and joint accountability, which are again based on the critical premise of empowering and engaging marginalized groups [67]. Communities are where the dynamics highlighted in the causal loop diagram play out and impact the health of people. Systems leadership enables distributed leadership, with communities able to create solutions that take into account systems dynamics at a local scale [68,69].

The causal loop diagram and system insights generated in this study will inform group model building (GMB) with a case study community. The intent of the GMB is to understand how the type of system structure identified here play out in practice within community. Results of this community level causal loop diagram can then inform discussions between community and government about COVID-19 response and recovery actions that will support positive and equitable health outcomes. The current study is an input into the community GMB, but a single input which presents one conceptualization of the system. As a rapid review intended to inform further group modelling work, there are limitations. The review considered articles between January-July 2020, restricted to English language and only four countries. Only the PubMed bibliographic database was utilized,

supplemented with Google searches for grey-literature. The causal loop diagram primarily reflects the New Zealand context, but does not attempt to go beyond high-level concepts and does not, for example, consider regional variations.

This qualitative causal loop diagram was developed to aid understanding of how SDH might contribute to health and equity outcomes related to COVID-19 response and recovery within New Zealand. The results demonstrate use of system insights to support COVID-19 response and recovery measures, as actions with short term impact could improve equity impacts of COVID-19 yet could at the same time reinforce pre-existing inequities. Systemic leadership could help shift system behavior towards longer term equity outcomes, by taking the opportunity to reset policies towards social justice and sustainability.

**Supplementary Materials:** The following are available online at https://www.mdpi.com/article/10.3390/systems9030052/s1, Figure S1: Study selection flowchart; Figure S2: Details of the reinforcing and balancing loops; Table S1: PRISMA checklist.

**Author Contributions:** S.S. and M.W. conceived the study. S.S. and S.M. analyzed the data and drafted the initial manuscript supervised by M.W. All authors critically reviewed and revised the initial manuscript. All authors have read and agreed to the published version of the manuscript.

**Funding:** This study is part of the Co-Search project funded by Health Research Council [HRC] and the Ministry of Health, New Zealand-(HRC 20/1066 [PI: Baker] & 20/990 [PI: Gray]).

**Data Availability Statement:** Not applicable.

**Acknowledgments:** We would like to acknowledge Amanda Kvalsvig (University of Otago) for feedback on the manuscript for further improvements.

**Conflicts of Interest:** The authors declare that there is no conflict of interest.

## Abbreviations

| | |
|---|---|
| Benefits | Unintended benefits (e.g., policy reset opportunities) |
| Capitalism | Capitalism-based development |
| Cases | COVID-19 cases |
| Colonization | Colonization and structural racism |
| Community response | Collective community response |
| Consequences | Unintended consequences (e.g., biodiversity loss, mental health) |
| Family support | Families needing social support |
| Govt support | Government economic support |
| Health access | Equitable access to health services |
| Health response | COVID-19 health responses (e.g., lockdown, border shutdown) |
| Indigeneity | Indigenous knowledge utilization |
| Infection risk | Infection risk among at risk groups |
| Innovation | Job and service delivery innovation |
| Leadership | Pandemic decision-making and leadership |
| Resources | Resources for the health and social response |
| SE impact | Socio-economic impact |
| SE inequities | Socio-economic inequities |
| Service design | Design of health and social services based on colonial/western worldview |

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
