# Peer review of "Social Determinants of Health Influencing the New Zealand COVID-19 Response and Recovery: A Scoping Review and Causal Loop Diagram"

_systems, doi:10.3390/systems9030052_

Round 1
Reviewer 1 Report
Topic addressed
This article addresses the topic of how certain aspects of a society influence the health of underprivileged groups in the population in the case of COVID19 in New Zealand. The work described is from an early phase of a larger project; it contains a qualitative system dynamics model expressed as a causal loop diagram and the authors describe how a scoping research has allowed to identify relevant literature and how it was analyzed to build the model. A preliminary analysis of the model in some interpretations in terms of the dynamic behavior of the problem are also included.
Strengths
Paper is when structured and well written. The introduction offers relevant information for readers to understand the relevance of the problem. The methods used are clearly identified and generally described in sufficient detail. The limitations of a qualitative model are stated, and it is clear that these limitations will be overcome in the remaining stages of the project.
Weaknesses
In my opinion, the structure of the qualitative model could be explained more clearly in the text, and I believe that the relationship between the examples given in the discussion section at the qualitative model could be stated more explicitly. This would probably allow readers who are interested in the field of social determinants of health to better understand the structure and the dynamic implications of the work described here, and maybe decide which aspects could be used for in their own work.
Discussion
I found the article very interesting, and I think is a good example of a cautious use of qualitative system dynamics work. My only concern is that some aspects of the qualitative model (the causal loop diagram and the treatment of its content in the text) could become clearer and easier to grasp. Therefore, I have organized on my comments as two sections the first section only refers to the presentation of the qualitative model, the second section contains some minor details.
Concerning the qualitative model
- In section 2.4, you introduce the key concepts and rules of causal loop diagraming. I had three remarks on this:
- On line three of the first paragraph, instead “tool that perhaps dynamic relationships between different elements”. A causal loop diagram (CLD) contains only the names of variables, causal links with the polarity and loop identifiers, and they all represent important aspects of the causal structure. CLDs do not display the behavior of the variables they contain. At the causal links’ polarities refer to those structural aspects that determine a variable’s reaction (change of behavior): I think these diagrams represent relationships. The developers of such a diagram and their users can use the diagram to mentally simulate or predict likely dynamics. But if you say “dynamic relationships”, I suspect readers will understand that the diagram directly displaced dynamics, which seems incorrect to me.
- In the same paragraph, you define the meaning of polarity using the abbreviated definition. This abbreviated definition has caused a lot of debate in the system dynamics community because it can easily lead to incorrect inferences. For example, “births ®+ population” seems to mean that when births decrease, population will also decrease - but this is not correct: when births decrease, population will be less than it would have been without the decrease of births. If you have doubts, you can find a discussion of these aspects can be found in (Schaffernicht, 2010), and otherwise there is an extensive technical introduction in Sterman’s (2000) chapter on causal loop diagrams.
- It may be hard to decide how acquainted the majority of readers are with causal loop diagrams. For those readers who are not familiar with this, it might be valuable to suggest some references in this section of your article.
- Table 1 is tremendously interesting. I only found that some of the “determinants” (variables?) Are hard to understand, and some variables appear in several or many rows. I thought it would be nice to have a list of variables which includes a brief definition of each of them. This can be in appendix the main text; it can even include the list of research papers where each variable is mentioned. I would have found that very helpful and interesting, and other readers might find so too.
- The causal loop diagram is complex, and I have a series of questions and suggestions. The diagram is the centerpiece of the article, and it is certainly important that readers can read it and absorbed information without avoidable difficulties:
- Are all the variables mentioned in table 1 also contained in the diagram?
- Color coding different components is certainly a good idea, but I have problems reading what is written in the dark blue and dark red variables.
- It may be impossible to of why is that some of the literature references are crushed by causal links, but maybe you could further reduce the number of these crossings.
- Feedback loops are not indicated by the traditional loop symbol; they may more salient if the symbol is used.
- Possibly, the overall layout and presentation of the diagram could be improved such as to minimize the crossing causal links and maybe use other tricks to increment clarity. As an interesting set of recommendations from George Richardson the following URL:
- https://onlinelibrary.wiley.com/pb-assets/assets/10991727/GPR_Diagram_improvement-1509468985000.pdf
- You seem to have created the diagram using the Vensim software, and readers who are also users of the same software package might find it nice if you share a copy of the file as supplementary material of the article.
- Six feedback loops are signaled in the diagram. Are these loops the only ones in the model, or did you select a subset of loops for the diagram? Do these loops interact, thereby creating a hierarchy of loops? This is an important consideration, because if two or more loops interact, it is not so easy to discuss them the text after the diagram in an isolated manner. Also, it is very hard to see what exactly each loop consists of. There are several ways how to help readers: you can develop a table one row for each loop and then mentioned the sick sequence of variables belonging to each loop. You can also offer an isolated causal diagram of each loop, which I would find quite great to accompany the text paragraphs discussing each of the loops.
- In the text paragraphs after the figure, I find the relationship between the causal loop diagram and text would be so much easier to understand if you somehow make the variables used in the text salient (for instance, by printing them in italics). Actually, readers might expect you it textile description of each sequence of variables and causal links, to make sure you provide the correct interpretation of what each loop means and does. This would also make visible how exactly you have to write the expected dynamics from the cost loop diagram (qualitative model) - especially if two or more of the feedback loops are interconnected.
Minor details
- Sometimes you write “New Zealand”, sometimes “Aotearoa New Zealand”. In the abstract, I first thought this is a region in New Zealand, because the title only has “New Zealand”. Then I started suspecting it might be the original name of the islands, looked it up on the Internet and now I know. You might spare other readers the doubt and the search by defining what it means the first time it appears. And maybe reading becomes easier by using always the same formulation.
- On p.2, line 3, “factors determinants” reads strange.
- 2, “However, an June 2020 survey […]”. I am not a native speaker, but I would bet it is “a June 2020 survey”.
- On p. 2, you write “Pasifika”, but in other places it is “Pacific communities”: do both terms refer to the same group?
- On p. 3, you describe the search terms for the PubMed database. There are several logical operators; the first one is “AND”, but the other ones are “and” and “or”. I would find it much easier to read if they all are in capital letters.
- 4, line 2: “impacts suggest”, the second part should not be in plural.
- 4, second paragraph: “casual“ loop diagram should be “causal”.
- 4, third paragraph: “which is also the intention of this study”. I found “intention” ambiguous: it’s Matt’s mean that this is what to you want to do or what you want to achieve, and in the first case I would wonder if you have done it.
- Table 1 is very informative but also difficult to read. I believe it might be a little easier if you represent the causal links by “®” instead of “-“, and you might even display each link’s polarity like “®+” and “®-”.
- On the paragraphs after figure 1, you explain essential features of the qualitative model represented by the causal loop diagram. I wonder why you refer to the feedback loops as “RL” and “BL”, if you have previously introduced readers to expect that the loops are symbolized by “R” and “B”. I also wondered if the text might become easier to read if you name the feedback loops referred to these names together with their identifier (R1 or B1 etcetera).
- Third paragraph of section 4: “key insights from this causal loop diagram more unsurprising”. If they were unsurprising, does that mean that you wasted your time developing it and that readers would only find information they already expected to find?
References mentioned in the review
Schaffernicht, M. F. (2010). Causal loop diagrams as means to improve the understanding of dynamic problems: a critical analysis. Systems Research and Behavioral Science, 27(6), 13.
Sterman, J. (2000). Business dynamics - Systems Thinking and Modelling for a Complex World: McGraw Hill.

Reviewer 2 Report
This paper read very well and was easy to digest. My first comment is that I could not find the reference [30] 'Arksey & O'Malley' - which is unfortunate because the study framework was adapted from this work.
I really liked the Introduction - it was very well presented and provided a good narrative. I have two main comments for the Introduction: (1) The aim seems weak - the stated aim is to create a CLD but I think you need to look more at the motivation for mapping out the system using systems thinking technique i.e. to conceptualise the complexity of the system that was described - the rationale (to aide understanding) also seems weak. (ii) The aspect of 'recovery' was not covered until very end (but appears in the title of this manuscript) - there needs to be context provided for this in the introduction.
In the Methods section, the missing reference of Arksey & O'Malley (at least from version that I downloaded) is a problem. There were a couple of instances where evidence was needed to substantiate a claim e.g. section 2.1 'the pandemic was predicted' but there is no evidence that there was a prediction.
RQ2 - addresses only 'health' but RQ1 addresses 'health and equity' - shouldn't they both cover 'health and equity'?
I feel that the subjective process outlined in section 2.3 needs to include a comment about potential biases in this process i.e. the notion that several potential causal chain linkages were suggested indicates this needs to be addressed in this section.
Section 2.4:
'conventions of system dynamics' needs a suitable reference. Same with the statement that 'the dominant loop is usually the one with the least delays'. Also in this section it is stated that the 'loops are generally balancing ... or reinforcing'. In terms of systems terminology, they are only balancing or reinforcing.
Results - my main comment here is that the CLD (Fig 1) is not easy to interpret. The variable names are small and it is very difficult to determine the feedback loops from the figure. I suggest a table that summarises the sequence of variables involved in each of the loops.
I am also not sure in the value of showing delays in your CLD - there are many delays shown but there is not a lot of text about the delays (right at the end of the Results it is mentioned and there is sparring reference to delays in the Discussion).
Discussion: My feeling is that the discussion addresses RQ2 more than RQ1?
You state 'Using systems perspectives' as a qualifier in the Discussion but I don't believe that you have used this perspective. Rather I believe that you have used some systems thinking tools. If you were taking a systems perspective then you need to consider explicitly the link between system structure and system behaviour - this includes the role of delays (which there are plenty in the CLD), which can cause overshoot, undershoot and oscillation behaviour.
Round 2
Reviewer 2 Report
I am happy with the authors comments to my review.